# Epidermal Growth Factor Receptor (EGFR) Gene Polymorphism May be a Modifier for Cadmium Kidney Toxicity

**DOI:** 10.3390/genes12101573

**Published:** 2021-10-02

**Authors:** Chun-Ting Lin, Ting-Hao Chen, Chen-Cheng Yang, Kuei-Hau Luo, Tzu-Hua Chen, Hung-Yi Chuang

**Affiliations:** 1Department of Public, College of Health Sciences, Kaohsiung Medical University, Kaohsiung 807, Taiwan; u105570013@kmu.edu.tw (C.-T.L.); u105570007@kmu.edu.tw (T.-H.C.); 2Departments of Occupational Medicine and Family Medicine, Kaohsiung Municipal Siaogang Hospital, and Kaohsiung Medical University, Kaohsiung 807, Taiwan; u106800001@kmu.edu.tw; 3Graduate Institute of Medicine, College of Medicine, Kaohsiung Medical University, Kaohsiung 807, Taiwan; u107800007@kmu.edu.tw; 4Department of Family Medicine, Kaohsiung Municipal Ta-Tung Hospital, and Kaohsiung Medical University, Kaohsiung 807, Taiwan; 980264@kmuh.org.tw; 5Department of Occupational and Environmental Medicine, Kaohsiung Medical University Hospital, Kaohsiung 807, Taiwan; 6Department of Public Health and Environmental Medicine, College of Medicine, and Research Center for Environmental Medicine, Kaohsiung Medical University, Kaohsiung 807, Taiwan

**Keywords:** cadmium, EGFR polymorphism, estimated glomerular filtration rate, kidney function

## Abstract

The results of many studies indicate that cadmium (Cd) exposure is harmful to humans, with the proximal tubule of the kidney being the main target of Cd accumulation and toxicity. Studies have also shown that Cd has the effect of activating the pathway of epidermal growth factor receptor (EGFR) signaling and cell growth. The EGFR is a family of transmembrane receptors, which are widely expressed in the human kidney. The aim of this study was to investigate the kidney function estimated glomerular filtration rate (eGFR), and its relationship with plasma Cd level and EGFR gene polymorphism. Using data from Academia Sinica Taiwan biobank, 489 subjects aged 30–70 years were analyzed. The demographic characteristics was determined from questionnaires, and biological sampling of urine and blood was determined from physical examination. Kidney function was assessed by the eGFR with CKD-EPI formula. Plasma Cd (ug/L) was measured by inductively coupled plasma mass spectrometry. A total of 97 single-nucleotide polymorphisms (SNPs) were identified in the EGFR on the Taiwan biobank chip, however 4 SNPs did not pass the quality control. Multiple regression analyses were performed to achieve the study aim. The mean (±SD) plasma Cd level of the study subjects was 0.02 (±0.008) ug/L. After adjusting for confounding variables, rs13244925 AA, rs6948867 AA, rs35891645 TT and rs6593214 AA types had higher eGFR (4.89 mL/min/1.73 m^2^ (*p* = 0.035), 5.54 mL/min/1.73 m^2^ (*p* = 0.03), 4.96 mL/min/1.73 m^2^ (*p* = 0.048) and 5.16 mL/min/1.73 m^2^ (*p* = 0.048), respectively). Plasma cadmium and rs845555 had an interactive effect on eGFR. In conclusion, EGFR polymorphisms could be modifiers of Cd kidney toxicity, in which rs13244925 AA, rs6948867 AA, rs35891645 TT and rs6593214 AA may be protective, and Cd interacting with rs845555 may affect kidney function.

## 1. Introduction

The kidneys are the primary organs related to toxic effects on the human body. The main nephrotoxic substances include heavy metals, antibiotics, and analgesics [1]. Long-term cadmium (Cd) exposure in the environment, which enters and accumulates in the human body through the lungs or gastrointestinal tract, has a great impact on the kidneys. The proximal tubules have the function of active absorption and secretion, with the S1 section being the main target that accumulates cadmium and induces toxicity [1,2]. Many countries have discovered from large-scale database research that cadmium exposure can affect kidney function. As the human body’s cadmium exposure becomes higher, the estimated glomerular filtration rate (eGFR) is significantly lower. Previous studies have shown that a blood Cd level of 0.6 μg/L or higher also shows an association with risks of developing chronic kidney disease (CKD, eGFR<60 mL/min/1.73 m^2^) [3,4,5,6,7].

EGFR (Epidermal growth factor receptor), a member of the transmembrane receptor family, which belongs to the subclass of the tyrosine kinase receptor superfamily, receives extracellular signals, activates the downstream pathways in the cell, and stimulates the intracellular response. The EGFR family has four members: epidermal growth factor receptor (HER1, EGFR, ErbB1); Neu / HER2 / ErbB2; HER3 / ErbB3 and HER4 / ErbB4 [8]. EGFR is widely present in the kidneys of mammals, including in the proximal convoluted tubules, cortical and medullary collecting ducts [9,10]. Many EGFR ligands have also been confirmed to exist in kidney cells, including TGF-α, EGF, HB-EGF..., etc. [11]. Among them, EGF is the most common ligand with the highest affinity found [12]. Activation of EGFR by its exogenous ligands, such as EGF, could enhance the recovery of renal function. However, some recent studies reported that EGFR activation also contributes to detriment of renal diseases in animal models such as obstructive nephropathy, diabetic nephropathy, hypertensive nephropathy, and glomerulonephritis. EGFR has genetic polymorphisms associated with glioma [13,14,15,16,17], lung cancer [18,19,20,21], Alzheimer’s disease [22], prostate cancer [23,24], and gastric cancer [25].

There are three downstream signaling pathways of EGFR, namely the extracellular signal-regulated kinase (ERK) pathway, the Janus kinase/signal transducers and activators of transcription (JAK/STAT) pathways, and the phosphoinositide 3 kinase (PI3K)/Akt pathways [8]. These pathways are related to cell survival, proliferation, dedifferentiation, and migration. There is evidence that cadmium can activate the PI3K/AKT pathway of EGFR [26]. Cadmium exposure can increase the signaling factors downstream of EGFR, and is related to the phosphorylation of AKT, thereby activating the PI3K/AKT signaling pathway [27]. Studies have also confirmed that cadmium can activate AKT, ERK1/2, and HIF 1 pathways through reactive oxygen species (ROS), which are all related to EGFR [28].

The literature reports that human epidermal growth factor (EGFR), cadmium, and renal function were correlated with each other, but researchers have not proposed the mechanism of correlation among them. Therefore, this study aimed to analyze the relationship between epidermal growth factor (EGFR), blood cadmium, and renal function through data and blood samples from the Taiwan biobank. The purpose of this study was to analyze the association between low-level cadmium exposure and EGFR gene polymorphism and eGFR.

This article aimed to study EGFR genotypes and the modification of the association between plasma cadmium and kidney function in a subgroup of the general population.

## 2. Materials and Methods

### 2.1. Taiwan Biobank

This study is a cross-sectional study. The research data were obtained from the Taiwan Biobank (TWB) of Academia Sinica, Taiwan. This research applied and obtained required data and various information after procedural, scientific, and ethical reviews. The admission process of the TWB database is divided into three parts, followed by physical examination, questionnaire interview, and specimen collection. The blood and urine of the subjects were entrusted to the Chang Gung Memorial Hospital Linkou General Medical Center for testing. The subjects were a group from the general population, for normal control against another studies in Taiwan (56% males and 44% females, aged 30 to 70 years). A total of 500 people were randomly selected by frequency matching according to age and gender. Participants in this study were healthy, and did not have any cancers, autoimmune diseases, or other catastrophic illness. The study was approved by the Institutional Review Board Kaohsiung Medical University Hospital (KMU-HIRB-E(I)-20150259, initial date of approval: 6 January 2016) and approval was waived for individual consent form, due to deidentification in Taiwan BioBank data and specimens.

TWB uses Whole Genome Sequencing, designed by the National Center for Genome Medicine (NCGM) in cooperation with Affymetrix, USA, and uses the Axiom Genome-Wide Array Plate chip system to select a total of 653,291 SNPs, called Taiwan Biobank chip, from which we searched for epidermal growth factor receptor (EGFR) genes, identifying a total of 97 EGFR sites (SNP) for use in this study. To test the quality of these 97 SNPs (QC test), the sample missing rate, Harbin balance, SNP missing rate, minor allele frequency, and heterogeneity rate were detected. The detailed steps are shown in Figure 1. A total of 93 SNPs eventually passed QC testing and were included in the research analysis.

### 2.2. Plasma Cadmium

The subjects were part of the working population of Taiwan (56% males and 44% females, aged 30 to 70 years), collected from TWB. A total of 500 people were randomly selected by frequency matching according to age and gender, and plasma cadmium analysis was performed with 5 mL plasma retained in TWB. The measurement was carried out in the laboratory of Kaohsiung Medical University by inductively coupled plasma mass spectrometry (ICP-MS, Thermo Scientific XSERIES 2).

During the sample preprocessing step, 1% nitric acid (15 mL of nitric acid was taken, 500 mL of deionized water was added slowly, and then deionized water was added to make it 1000 mL) was added to the plasma sample, a 1:10 dilution was carried out, and left for 10 min. ICP-MS standard solution (Accu Standard, MES 04-1) was used for the configuration of a calibration curve, which was diluted with concentrations of 0.1, 0.2, 0.5, 1, 2, 5, 10, 20, 50, 100, 200, 500, 1000, 2000, 3000 ug/L, and the elements following this calibration curve showed a linear relationship (r > 0.995).

Before analyzing unknown concentrations, quality control (QC) and quality assurance (QA) analyses were performed to ensure the accuracy and precision of the experiment. QC ensures the stability of equipment and systems through three repeated tests of standard reference materials (SRM), and the coefficient of variation (CV) must be less than 3%. QA analysis is mainly to conduct a random analysis of standard reference materials. Each analysis result is put into the calibration curve and meets between 90% and 110%.

### 2.3. eGFR (Estimated Glomerular Filtration Rate) Equation

In recent years, many European and American studies have recommended the use of the CKD EPI formula instead of the MDRD formula. Studies have pointed out that the CKD EPI formula is more accurate than MDRD when estimating relatively high glomerular filtration rates, and the deviation is relatively small [29]. It can also reduce the overdiagnosis of CKD [30]. Previous studies have indicated that MDRD is relatively inaccurate in estimating a glomerular filtration rate greater than 60 mL/min/1.73 m^2^, and thus this study used the international version of the CKD EPI formula for the estimation of glomerular filtration rate.

### 2.4. Statistical Analysis

Continuous variables were represented by the mean value (±standard deviation), and categorical variables were represented by the number (percentage). To obtain statistical power, genotypes with less than 20 people were merged with another genotype. Take rs763317 (SNP 4) as an example: The genotypes and numbers of rs763317 (SNP 4) were AA in 18 people, AG in 183 people, and GG in 288 people, which were merged into two groups, namely: 201 people with AA+AG and 288 people with GG. The regression diagnosis was used to ensure the accuracy of the analysis results (Table 1); the generalized linear regression model was used to explore the relationship between plasma cadmium, the EGFR gene polymorphism, and eGFR after controlling for interference factors; finally, the interaction of plasma cadmium and SNPs was added to the generalized linear regression model to explore the relationship between plasma cadmium, the EGFR polymorphism, and eGFR after controlling for interference factors. All statistical analyses were carried out with SAS 9.4. A two-tailed *p*-value < 0.05 was considered significant.

## 3. Results

### 3.1. Characteristic and Laboratory Information

The average (±standard deviation) age in this study was 48 (±10.9) years old, of which males were 47.5 (±11.0) and females were 49.3 (±10.7) years old; the ratios of males and females were 55.83% and 44.17%; the average (±standard deviation) body-mass index (BMI) was 24.4 (±3.4), of which the average (±standard deviation) BMI for men was 25.1 (±3.2) and the average (±standard deviation) BMI for women was 23.6 (±3.6). The drinking status was divided into “currently drinking” and “currently not drinking”, the ratios were 10.63% and 89.37%; the smoking status was divided into “currently smoking” and “currently not smoking”, the ratios were 11.04% and 88.96%. 41 (8.38%) males and 14 (2.86%) females had hypertension; 17 males (3.47%) and 10 females (2.04%) suffered from diabetes; 25 males (5.11%) and 11 females (2.24%) had kidney stones. The basic data of the demographic characteristics in the study are shown in Table 1.

As shown in Table 2, each biochemical measurement value of this study was expressed as an average value (±standard deviation). The cadmium in plasma was 0.022 (±0.008) µg/L; the creatinine in blood was 0.777 (±0.194); blood urea nitrogen (BUN) was 13.046 (±3.487); albumin was 4.615 (±0.249); international CKD-EPI was 101.372 (±14.281).

### 3.2. SNPs Information

The SNPs of 93 EGFRs are shown in the Appendix A. Among them, 9 sites did not comply with Harvin’s law, namely: rs6965469 (SNP1) (*p* = 0.042), rs13234622 (SNP32) (*p* = 0.016), rs2072454 (SNP33) (*p* = 0.010), rs2075110 (SNP34) (*p* = 0.021), rs4947986 (SNP35) (*p* = 0.021), rs845551 (SNP42) (*p* = 0.012), rs9642393 (SNP46) (*p* = 0.002), rs845555 (SNP47) (*p* = 0.045), rs940806 (SNP67) (*p* = 0.038). There were two SNPs with minor allele frequencies (MAF less than 5%): rs10234806 (SNP10) (*p* = 0.047) and rs11979255 (SNP15) (*p* = 0.049).

### 3.3. Association between Plasma Cadmium, EGFR and eGFR

As shown in Table 3 (Model 1), after adjusting the interference factors, there was no significant difference between plasma cadmium concentration and kidney function (β = −134.93, *p* = 0.08), for every 1 ug/L increase in plasma cadmium, the glomerular filtration rate decreased by 134.93 mL/min/1.73 m^2^. After entering the 93 SNPs of EGFR into the regression model one by one and adjusting the interference factors, the results showed that rs13244925 AA (β = 4.89, *p* = 0.035), rs6948867 AA (β = 5.54, *p* = 0.030), rs35891645 TT (β = −4.96, *p* = 0.048), and rs6593214 AA (β = 5.16, *p* = 0.042) were significantly related to the glomerular filtration rate.

An interaction term was added to the multivariate logistic regression to analyze the relationship between plasma cadmium and the EGFR gene and renal function (Table 3, Model 6). Among the 93 SNPs, only rs845555 CT genotype interacted with plasma cadmium. For every 1 ug/L increase in the plasma cadmium concentration of people with rs845555 CT genotype, the eGFR only decreased by 0.22 mL/min/1.73 m^2^ (*p* = 0.02), while for every 1 ug/L increase in the plasma cadmium concentration of people with CC and TT types, the eGFR decreased by 440.02 mL/min/1.73 m^2^ (*p* = 0.01). Plasma cadmium and rs845555 had an interactive effect on eGFR at the same time. When the plasma cadmium concentration was higher than 0.020 ug/L, the eGFR of people with rs845555 CC and TT genotypes greatly decreased with the increase in plasma cadmium concentration, while this did not happen in people with rs845555 CT genotype (Figure 2). These regression lines could be extended to plasma Cd concentrations of 0.2–0.3 ug/L (still within the normal range), then the individuals with CC and TT types would have significantly lower CKD-EPI values and greatly impaired renal function.

## 4. Discussion

The results of the present study show that after adjusting for interference factors, plasma cadmium was negatively correlated with eGFR, but there was no statistically significant relationship (*p* = 0.08); the reason might that be the sample numbers were not large enough. Additional studies with increased sample size may find a statistically significant association between plasma cadmium levels and lower eGFR. However, in people with some genotypes, such as rs35891645 (β = −154.92, *p* = 0.044) and rs6593214 (β = −155.78, *p* = 0.042), the negative effect of cadmium on renal function became statistically significant, which showed that in the low-exposure group (normal group), cadmium may be particularly harmful to kidney function for people with the above two genotypes.

This is a pioneering study on human EGFR, plasma cadmium concentration, and eGFR. This study found that among 97 SNPs in Taiwan Biobank, rs13244925 AA genotype (β = 4.89) and rs6948867 AA genotype (β = 5.54), rs35891645 TT genotype (β = 4.96), and rs6593214 AA genotype (β = 5.16) significantly affected kidney function, and the heterozygosities of the above four genes were not significant, which may suggest that they are protective genes. However, because the effect was not high, instead of exploring the protective effect caused by EGFR gene polymorphism, it may be quicker and more effective to directly reduce the exposure of cadmium in the industrial environment.

There are few studies on the above four genes, and no studies have shown that they are related to any disease. After adjusting the plasma cadmium, the AA genotype of rs13244925 was significantly related to eGFR. This is the first study to find that rs13244925 is related to kidney function. Even though the effect on kidney function is small, this study is the first to discover that rs6948867 AA genotype, rs35891645 TT genotype, and rs6593214 AA genotype are related to kidney function.

Plasma cadmium concentration and rs845555 have an interactive effect on eGFR. People with rs845555 CT genotype experience a reduced effect of cadmium on eGFR. When the plasma cadmium concentration increased by 1 ug/L, the eGFR of people with rs845555 CT genotype only decreased 0.22 mL/min./1.73 m^2^ (*p* = 0.02), while the eGFR of people with CC and TT genotypes decreased by 440.02 mL/min/1.73 m^2^, so rs845555 CT genotype may be an overdominant, protective gene [31]

According to the recommended guidelines for special health examination management, the abnormal result is that the concentration of cadmium in the blood is greater than 5μg/L, and the average plasma cadmium (±standard deviation) of this study group is 0.022(±0.008) μg/L, and the geometric mean is 0.021 μg/L. In the present study, it was found that cadmium exposure did not display a statistically significant correlation with eGFR, which may be due to the average blood cadmium concentration of the population in this study being relatively low. However, plasma cadmium has a certain effect on the β coefficient of kidney function.

There are some limitations to this study: First, the human epidermal growth factor receptor (EGFR) has 1505 known sites in NCBI, but only 97 sites in Taiwan Biobank v1.0. However, among the 97 sites in the chip, we have found 5 SNPs that are significantly related to kidney function. If the chip had more sites for analysis and research in the future, we could find more EGFR sites related to kidney function. Second, this study is a cross-sectional study. Although the causal relationship between plasma cadmium and eGFR cannot be judged from the data, it has been demonstrated in many other studies that there is indeed a correlation between the two. Third, although most studies on cadmium exposure use cadmium in whole blood, only cadmium in plasma is active. Therefore, in the study of toxic effects, cadmium in plasma is more representative than cadmium in whole blood. Fourth, we did not have the data of drugs that might affect kidney function. However, the participants were a group of general population for normal control to another studies. Subjects in this study were healthy, and the uses of drugs affecting kidney function might be low. Finally, this study used the internationally accepted CKD EPI formula to estimate the glomerular filtration rate, and some scholars in Taiwan have also proposed an adjusted Taiwan version of the CKD EPI formula [32]. This study compared the correlation coefficients between the Taiwan version of CKD EPI and the international CKD EPI formula and found a high correlation (Pearson correlation = 0.99, *p* < 0.0001). This study also used the Taiwan version of CKD-EPI formula to estimate the glomerular filtration rate and found that the SNPs with a significant difference were the same as the SNPs analyzed using the international version of CKD-EPI, and there was no significant difference in the β coefficient. Finally, this research used the international version of CKD-EPI for future comparison with other documents.

## 5. Conclusions

The present study found that 4 SNPs in human epidermal growth factor receptor (EGFR) are related to the estimated glomerular filtration rate (eGFR); however, they (rs13244925, rs6948867, rs35891645, rs6593214) are not exons or introns, they have unknown function. Thus, we suggest they could be included in future studies. The CT genotype of rs845555 and plasma cadmium have an interactive effect on eGFR, which may be an overdominant protective gene. The rs845555 is an intron location. Its protective role against cadmium could be investigated in the future studies.

## Figures and Tables

**Figure 1 genes-12-01573-f001:**
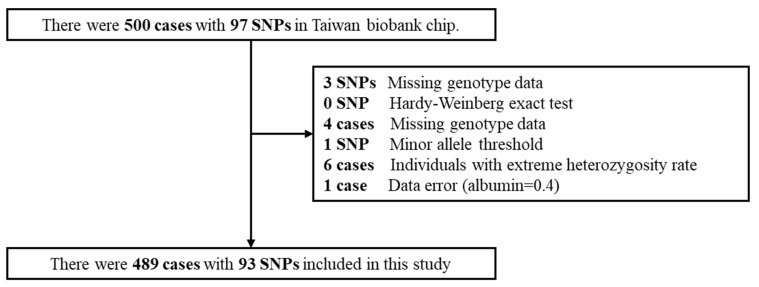
The selections of participants and SNPs in Taiwan Biobank.

**Figure 2 genes-12-01573-f002:**
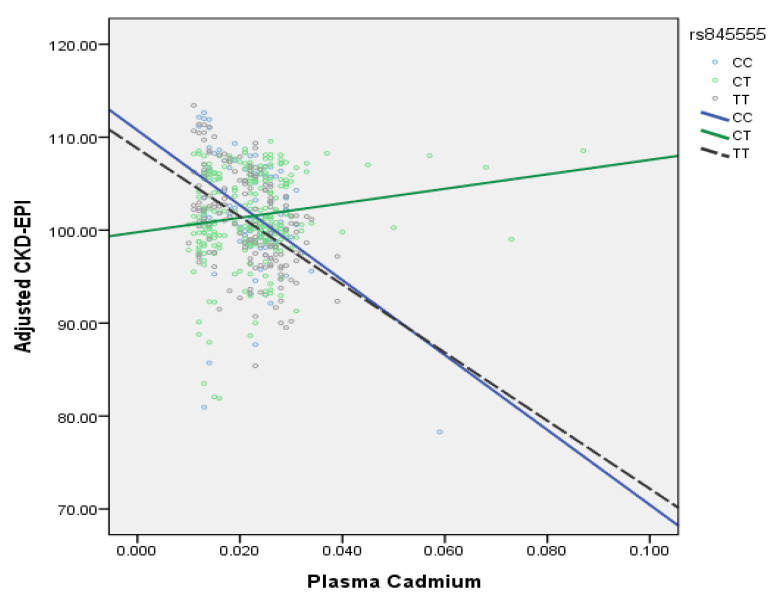
Interaction plot for Cadmium-rs845555 polymorphism and adjusted CKD-EPI.

**Table 1 genes-12-01573-t001:** Demographic and clinical characteristics of participants.

Variables		*n* or Mean *
Participants		489
	male	273 (55.8%)
	female	216 (44.2%)
Age		48.3
	male	47.5 ± 11.0
	female	49.3 ± 10.7
Weight		66.9 ± 12.2
	male	73.2 ± 10.5
	female	58.9 ± 9.1
Height		165.1 ± 8.6
	male	170.8 ± 5.9
	female	157.9 ± 5.4
Body-mass index, BMI		24.4 ± 3.4
	male	25.1 ± 3.2
	female	23.6 ± 3.6
Alcohol		
	never or quit	437 (89.4%)
	current	52 (10.6%)
Smoking status		
	never or quit	435 (88.9%)
	current	54 (11.0%)
Hypertension		55 (11.2%)
	male	41 (8.4%)
	female	14 (2.9%)
Diabetes		27 (5.5%)
	male	17 (3.5%)
	female	10 (2.0%)
Kidney stone		36 (7.4%)
	male	25 (5.1%)
	female	11 (2.2%)

* for the continuous variables, mean ± standard deviation was presented.

**Table 2 genes-12-01573-t002:** Biochemical indicators of participants.

Variables		Mean	Standard Deviation	Median	IQR *
Cadmium(ug/L)		0.022	0.008	0.022	0.011
	male	0.021	0.007	0.022	0.012
	female	0.023	0.009	0.022	0.010
BUN(mg/dL)		13.05	3.49	12.60	4.70
	male	13.59	3.40	13.00	4.80
	female	12.36	3.48	12.00	4.65
Creatinine(mg/dL)		0.78	0.19	0.62	0.79
	male	0.90	0.15	0.89	0.15
	female	0.62	0.11	0.61	0.13
CKD-EPI ^a^		101.37	14.28	102.74	18.06
	male	98.21	14.40	99.50	18.12
	female	105.37	13.11	106.12	18.39

^a^ CKDEPI equation = 141 × min (SCr/κ, 1)α × max(SCr/κ, 1)−1.209 × 0.993Age × 1.018 (if female); * IQR: interquartile range.

**Table 3 genes-12-01573-t003:** Selected general linear regression models of CKD-EPI associated with plasma cadmium and SNP types.

Variables/Model	Model 1		Model 2		Model 3		Model 4		Model 5		Model 6	
	β	S.E.	β	S.E.	β	S.E.	β	S.E.	β	S.E.	β	S.E.
Sex (Male/Female)	−6.08	1.34 *	−6.21	1.33 *	−5.83	1.34 *	−5.96	1.34 *	−5.95	1.34 *	−6.08	1.33 *
Hypertension (Yes/No)	−5.89	2.00 *	−6.18	2.01 *	−6.24	2.02 *	−6.19	2.02 *	−6.68	2.04 *	−5.89	2.00 *
Diabetes (Yes/No)	−12.11	4.44 *	−12.54	4.43 *	−11.83	4.44 *	−11.82	4.44 *	−9.58	4.67 *	−12.43	4.45 *
Kidney stone (Yes/No)	−1.41	2.35	−1.63	2.35	−1.64	2.35	−1.53	2.35	−1.54	2.35	−1.72	2.37
Smoking	2.24	2.05	2.32	2.05	2.01	2.07	2.15	2.05	1.81	2.06	2.16	2.05
drinking	−0.89	2.09	−1	2.09	−1.02	2.09	−0.97	2.09	−0.84	2.09	−0.91	2.09
BMI	−0.48	0.18 *	−0.44	0.18 *	−0.46	0.18 *	−0.45	0.18 *	−0.47	0.18 *	−0.45	0.18 *
Cadmium	−134.93	76.3 ^a^	−128.32	76.18	−149.75	77.03 ^b^	−154.92	76.77 *	−155.78	76.62 *	−440.02	167.44 *
rs13244925 (AA/CC)	-	-	4.89	2.31 *	-	-	-	-	-	-		
rs13244925 (AC/CC)	-	-	−0.65	1.29	-	-	-	-	-	-		
rs6948867 (AA/GG)	-	-	-	-	5.54	2.54 *	-	-	-	-		
rs6948867 (AG/GG)	-	-	-	-	1.46	1.32	-	-	-	-		
rs35891645 (TC/CC)	-	-	-	-	-	-	1.34	1.32	-	-		
rs35891645 (TT/CC)	-	-	-	-	-	-	4.96	2.5 *	-	-		
rs6593214 (AA/TT)	-	-	-	-	-	-	-	-	5.16	2.61 *		
rs6593214 (AT/TT)	-	-	-	-	-	-	-	-	1.53	1.32		
rs845555 (CT/TT)											−9.14	4.33 *
rs845555 (CC/TT)											1.17	6.34
Cd*rs845555-CT											448.94	190.89 *
Cd*rs845555-CC											3.42	274.06
intercept	120.21	4.82	119.27	4.89	119.21	4.88	119.22	4.89	119.63	4.89	125.57	5.82
R-square	0.120		0.132		0.13		0.129		0.128		0.135	

* *p* < 0.05; ^a^
*p* = 0.08; ^b^
*p* = 0.052.

## Data Availability

Restrictions apply to the availability of these data. Data was obtained from Taiwan Biobank (TWB) of Academia Sinica and are available [https://www.twbiobank.org.tw] with the permission of TWB.

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
