# Peer review of "Epidermal Growth Factor Receptor (EGFR) Gene Polymorphism May be a Modifier for Cadmium Kidney Toxicity"

_genes, 2021, doi:10.3390/genes12101573_

Round 1
Reviewer 1 Report
The manuscript titled “The epidermal growth factor receptor (EGFR) Gene polymorphism may be a modifier for cadmium kidney toxicity” by Lin Et al., describes the use of deidentified human samples to identify EGFR polymorphisms that are related to eGFR; with one of the polymorphisms demonstrating an interactive effect with plasma cadmium that may be protective.
- The manuscript has many grammatical and syntax issues. The authors should have a colleague who is proficient in English writing review the manuscript to improve on these issues.
- The introduction should have more information on cadmium-induced nephrotoxicity and the effects of cadmium on GFR and the proximal tubule epithelial cells. What levels of cadmium affect GFR and results in renal toxicity? How does this relate to the patients in your study?
- Were the patients screened for the use of other drugs that might affect kidney function?
- The discussion should contain more information from published studies on the protective effect or detrimental effect of EGFR activation on kidney function. This would set the stage for then discussing the polymorphisms and their observed roles as found in the study.
- What is the effect of the 4 EGFR polymorphisms on receptor signaling? Is receptor activation and signaling augmented or attenuated? At a minimum this could be included in future studies at the end of the conclusion section as the authors don’t mention future studies for this area of research.
- The authors should overexpress the CT genotype of rs845555 in vitro to determine if there is a protective effect against cadmium at the cellular level. At a minimum this could be included in future studies at the end of the conclusion section as the authors don’t mention future studies for this area of research.
- The conclusion does not really mention the big picture of how this information could be used to improve human health. What is the big picture?
Reviewer 2 Report
This study investigated the relationships between plasma Cd concentration, renal function, and SNPs in the EGFR gene using Biobank samples in Taiwan. The subjects' plasma Cd concentrations were very low, and the levels of CKD-EPI, a modified way of calculating eGFR, were in the normal range. The authors concluded that some SNPs in the EGFR gene might be potential modifiers of Cd-induced nephrotoxicity.
However, this study is problematic even from the starting point of hypothesis setting. EGFR and its downstream signaling pathways play multiple roles, mainly in the carcinogenic processes rather than renal dysfunctions. Carcinogenic effects of Cd have been found in workers exposed to high concentrations of Cd fumes and dust, not the general population. If the authors want to investigate the genetic polymorphisms related to Cd-associated renal dysfunction using this Biobank, they could have focused on the other more critical genes associated with Cd deposition, transport, or detoxification. The reason the authors selected the EGFR gene in this study was not fully explained and is beyond the reviewer's understanding.
Because Cd causes renal damage by impairing the reabsorption of biomolecules at the proximal tubules, urinary excretion of β2-microglobulin, a sensitive biomarker for Cd-induced defect in reabsorption, rather than the eGFR should be determined. The authors cited some papers on the relationship between Cd exposure and eGFR. However, these studies showed no correlation between these two indicators among the individuals whose Cd exposure is very low. The eGFR decreases only in the individuals highly exposed to Cd. The variation of eGFR values among normal individuals may be influenced by factors other than Cd exposure.
The authors showed that the regression line of the CT-type SNP (rs845555) of the EGFR gene is distinct from that of CC- or TT-type (Fig. 1). However, these regression lines, especially those for CC- and TT-types, are misleading. The maximum plasma Cd level was less than 0.1 ppm, indicating that these subjects’ Cd levels are well within the normal range. Suppose that the regression lines for the CC and TT types were correct. Then, if these regression lines were extended to plasma Cd concentrations of 0.2-0.3 ppm (still within the normal range), the individuals with CC and TT types would have significantly lower CKD-EPI values and greatly impaired renal function.
Furthermore, there were only 8-10 CT-type individuals in the range of 0.04-0.1 ppm plasma Cd, and their CKD-EPI values were within the normal range. On the other hand, only one CC-type individual was found in the same Cd range, and this individual happed to show a lower CKD-EPI value. These too few points might have significantly affected the regression of each SNP type.
Thus, it is fundamentally unreasonable to argue whether SNPs in EGFR alter the relationship between Cd exposure and renal function when blood Cd levels are in the normal range and renal damage is almost non-existent.
Minor points
Line 69.
The meaning of “two-by-two correlations” is not understandable.
Line 78.
The authors mentioned that this study is a “cross-sectional cohort” study, but it appears that the study setting is simply a cross-sectional study.
Reviewer 3 Report
This manuscript well demonstrated the possible involvement of EGFR polymorphism in Cd kidney toxicity. Although Cd has been known to be a toxic heavy metal for a long time, the target gene of Cd toxicity is remaining for elucidating. Authors well analyzed sufficient participants. Also, the statistical analysis was very meaningful. This manuscript is very useful for further research about gene polymorphism, kidney function, and Cd renal toxicity.
Reviewer 4 Report
This is a nice study examining the associations between eGFR, EGFR polymorphism and cadmium exposure as measured by plasma samples.
Only minor edits are suggested.
English grammar and usage need to be checked such as in line 149 “… had diabetic;”
Table 1; in the table heading, define the meaning of the numbers that follow “±”.
Line 194; while it is correct to say no significant difference is present with a p value of 0.08. The authors may consider inserting a statement that with an increase in N value, additional studies may find a statistically significant association between plasma cadmium levels and lower eGFR.
